# Autism, thy name is man: Exploring implicit and explicit gender bias in autism perceptions

**Rae Brickhill**[ID]**, Gray Atherton**[ID]**\*, Andrea Piovesan**[ID]**, Liam Cross**

Department of Psychology, Edge Hill University, Liverpool, United Kingdom

\* Athertgr@edgehill.ac.uk

**Data Availability Statement:** The data underlying the results presented in the study are available on the OSF, from: via https://osf.io/t3z2n.

**Funding:** The authors received no specific funding for this work.

## Abstract

Males are around three times more likely to possess an autism diagnosis than females. For years this was explained by accounts that considered the male gender more compatible with the autistic phenotype. However, new research suggests that a lack of understanding and recognition of the female autistic phenotype, and a predisposition to associate males with autistic traits, could lead to structural inequalities that hinder the identification of autistic females. To explore how autism and gender are more widely perceived, the present study tested implicit and explicit associations between autism and binary gender using the Implicit Association Test (IAT) and the Autism Quotient (AQ) presented alongside a male or female vignette. A significant association was found on the IAT, identifying an implicit bias towards males and autistic traits. The vignette AQ pairing also revealed some specific items perceived as explicitly male traits, while only reverse-scored items were perceived as female. These findings suggest that current perceptions and even metrics of autism are skewed towards males, which may hinder the identification and understanding of the female autistic phenotype.

## Introduction

Autism Spectrum Condition (ASC) is a complex neurodevelopmental condition characterized by impairments in communication and social interaction [1]. While this neurodevelopmental disorder has been recognized by since the 1940s [2], autism research has seen drastic growth in the past several decades due to autism's increasing prevalence [3]. In line with other neurodevelopmental conditions, autism is more commonly diagnosed in males than females [4], with three times more males diagnosed than females [5]. Females are also frequently diagnosed later in life than their male peers [6] and have an increased risk of remaining undiagnosed [7, 8]. More recently, researchers found that females had a higher average age of diagnosis than males and were more likely to be diagnosed in adulthood rather than childhood in contrast to males [9].

This later diagnosis of females has led some to question whether autism is perceived as a 'male' condition at its most basic level and thus leads to systematic discrimination against females who are autistic. This is an issue often raised by stakeholders in the autistic community, including parents of autistic female children [10], women who received a diagnosis later in life [7], and autistic adults with experiences of gender dysphoria [11].

**Competing interests:** The authors have declared that no competing interests exist.

## Gender stereotypes

Research suggests that behaviours and perceptions of men and women are governed by stereotypes, as they provide prompt assumptions toward members of social groups [12]. For example, males are perceived to have more agentic characteristics than females, as they are perceived to be more independent, assertive, and dominant; while increased empathy and greater communication abilities are often more widely perceived in females [13–15]. Females have also been found to be more emotionally expressive than males while being better able to demonstrate greater use of non-verbal communication both in-person and during online interactions [16, 17]. Shyness is also a common trait associated with females; shyness for example is much less likely to raise concerns for girls in school than boys due to the social expectation that girls are shy [18]. These expectations can then influence individuals' perceptions of men and women and how they act [19, 20]. This isc orroborated by comparing the pressure felt by adolescents to engage in same-gender behaviour, extending the current understanding of gender identity, as men reported pressure to avoid feminine-typed behaviour [21]. On average, females are also believed to be more susceptible to social pressures than males [22].

## Gender stereotypes and autism

Female stereotyped behaviours, in many ways, stand in stark contrast to typically recognized autistic traits. The language used to describe ASC is littered with certain gender expectations. For instance, the general view that women display more heightened emotional expressions, alongside greater motivation to communicate for social reasons than men, directly contrasts characteristics associated with autism [23]. Such biases may contribute to the systematic constraints that prevent girls from obtaining recognition for their ASC from their parents, educators, and peers at the same rate as their male counterparts. A reluctance to view a female as autistic likely has a detrimental effect on wellbeing. Improved understanding and more positive perceptions from peers are essential outcomes of a recognized diagnosis [24].

Recognizing an individual as autistic is critical for improving double empathy between the autistic person and the neurotypical community [25]. A public-facing autistic identity allows neurotypicals to recognize that an autistic person possesses a neurodivergent perspective equally deserving of understanding and respect. Failure to acknowledge and validate neurodivergent views is known as the double empathy problem, which may be compounded in autistic females. Specifically, as the stereotypes associated with females may be particularly at odds with perceptions of autism, there runs the risk of a heightened misunderstanding of neurodivergence in females. For instance, qualitative research [7] revealed how healthcare providers consistently dismissed autistic female clients' accounts of their autistic traits as more indicative of conditions such as depression, as these are none to have higher female ratios. Autistic participants have also discussed how their experiences of gender dysphoria were minimized or explained away by their autistic status [11]. These accounts show how stereotypes of neurodivergence and gender can reduce double empathy and inhibit the acceptance of autistic people who may not fit the typical male-autistic presentation.

There are several dominating accounts of the social features of autism that are at odds with female stereotypes. The theory of mind account [26], which cites autistic social differences as stemming from poor perspective-taking, may also reflect a gendered stereotype favouring males. Females are believed to be better able to take other's perspectives into account [27]. Indeed, theory of mind (ToM) tasks often reveal a female advantage, as assessed using both behavioural [28] and neural methods of investigation [29]. Another popular theoretical account of the social features of autism, the social motivation account [30], purports that

autistic people are less interested in social stimuli than neurotypicals. Similarly, research suggests that females tend to be more socially motivated than males [31, 32].

Interestingly, research has not found a female advantage in social tasks. For instance, autistic females appear to have similar difficulties reading emotions in their eyes as autistic males, despite an advantage on the same task in neurotypical females compared to males [33]. To minimize differences in ways that match society's expectations, autistic individuals often camouflage their traits [34] by attempting to emulate the social interaction style of what is deemed as neurotypical [35] to fit in more with their peers [36]. Motivation for camouflaging and masking social difficulties is increased by social pressures and expectations experienced by females [35]. For instance, women with autism 'pretend to be normal' by copying others' speech patterns and body language [7, 37] as a reaction to the societal messages they receive, which tell them as females they should be more sociable than males [10].

There is a worrying association between camouflaging, autistic traits, and diminished mental health [38], alongside stress associated with autism being 'found out' [39]. These additional gendered pressures are of pragmatic concern. One way to improve these outcomes for women on the spectrum is to understand and recognize ASC in females, particularly as it concerns early diagnosis [9]. Equal identification of female autistic traits is imperative as, among other benefits, those recognized as autistic experience greater acceptance as there is a reason and explanation for expressing atypical behaviour [24].

Understanding discrepancies in male and female identification rates is particularly imperative in the early years. Research suggests early intervention has many benefits, including more significant gains in communication and wellbeing [40]. Unfortunately, systematic male biases negatively impact the early identification of female autistic children. For instance, researchers found gender-related assumptions about male and female children's social engagement and activities with and without ASC at school [41]. Specifically, the social challenges of boys with ASC were viewed as more obvious than those of girls with ASC. This correlates with research that showed educators within a school setting expressed a reduced likelihood of seeking support for a female believed to have autism compared to an equivalent male [42]. Finally, parents' perception of their child's behaviours and difficulties may also affect an individual's motivation to camouflage due to social expectations varying depending on the child's gender [43]. This poses severe consequences for the welfare of females with ASC, putting them at a disadvantage to males in both educational and domestic contexts [6].

Another consequence of gendered stereotypes about autism is that they fuel common misconceptions of autism within the general population. Such stereotypes can result in unfavourable inferences about individuals with ASC based on their social, behavioural, and communicative differences, harming their mental health [44]. For instance, people form a more negative first impression of individuals with ASC than those without, whilst also being influenced by the individual's gender, with significant differences apparent between the judgements of autistic females compared to non-autistic females [44]. This bias has also been found to occur at an unconscious, implicit level through an Implicit Association Test (IAT) [45]. In an IAT, individuals are timed on their ability to pair a set of stimulus words with a similar meaning (i.e. 'good,' 'nice,' 'kind') with a categorical word (i.e. 'Muslim', 'British'). In an IAT test looking to test the association between autism and positive and negative words, a significant negative implicit attitudes towards autistic adults in the general population was found [46], despite participants asserting positive explicit attitudes. These results provide further explanations for the discrimination against autistic adults within society. Given the stereotypical associations between female traits and female sociality, discrimination may be particularly heightened towards autistic females.

Gendered stereotypes towards autism are also reinforced within the media as stereotypes and stigmas surrounding autism can be strengthened [47]. One example is the stereotypical use of male characters in *The Curious Incident of the Dog in the Night-Time*, *Rain Man*, *The Big Bang Theory* and *Atypical*. These popular representations can influence generalized perceptions of neurodevelopmental disorders, such as autism, even when an individual has personal experience with autism [48]. While media representations have been shown to affect implicit associations in general [49], this is especially important to consider for those that may not have access to personal experience, suggesting further reliance on the media to understand autism, further contributing to stereotyped attitudes. Furthermore, researchers report the representation of only "very high levels" to "extreme levels" of autism-related symptoms within the media [48]. This portrayal may further contribute to the high levels of missed diagnosis of females with less immediately apparent areas of functional need. Thus, the general public may form a highly stereotypical view of individuals with ASC, often through the lens of male rather than female characters.

## Gender biases in autism research

Gender bias has likely filtered into academic research, resulting in a lack of sensitivity towards the symptomology presented by ASC females [8], as the current diagnostic tools for autism are believed to be geared towards the male autism phenotype [50]. Some of this may stem from the fact that autism research favours male over female participants [8]. This sampling bias likely contributes to why male presentations of autism are often used as a guide for designing and implementing current diagnostic tools, exacerbating the misidentification of autistic females [51]. These challenges may explain why females with ASC must present with more functional needs (such as more concurrent behavioural, developmental, or mental health issues) to be diagnosed compared to males [52, 53]. Indeed, among studies with autistic individuals without intellectual disabilities, the ratio between males to females is as high as 10:1 [54].

Some suggest that there is one specific theory of autism that particularly perpetuates male autism stereotypes and lacks sensitivity towards the female autism phenotype [55]. The 'extreme male brain' theory of autism suggests that due to their systemizing nature and hindered emotional responses [56], autistic people are an example of a hyper-masculinized group of individuals regardless of gender. Some emphasise that the term 'extreme male brain' simply predicts that individuals with autism will show a masculinized pattern of scores of reduced empathy and above average systemizing rather than possessing an extreme of all masculine characteristics (such as aggression) [57]. However, the gendered term may affect how the general population views autism and associates it with males.

Some [58] argue that within the term 'extreme male brain', the use of 'brain' has replaced 'behaviour' with no explanatory power, causing confusion and little descriptive utility. Further questioning this term, researchers compared the AQ scores of neurotypical males and females with those of autistic males and females [3]. They found that autistic female AQ scores were only similar to those of males with ASC and not neurotypical males. This suggests that the scores produced by both males and females with ASC result from the neurodevelopmental condition itself rather than possessing a 'male brain.' A popularisation of this term can profoundly impact societal perceptions of autism in males and females by encouraging a gendered understanding of the disorder. Indeed, qualitative accounts of autistic people with gender dysphoria have specifically mentioned how such statements exacerbate their feelings of identity confusion [11].

The high ratio of males to females diagnosed with autism has led to a great deal of genetic research investigating whether there are sex-linked genes that account for autistic

symptomology. Along with the male brain theory, other influential theories, such as the female protective effect (FPE), suggests a complex differential liability model of autism and gender, which says that males carry more genetic risk factors for autism while females have specific protective factors that inhibit the development of autistic traits [59]. Similarly, some [60] hypothesize that males have a decreased plastic response to environmental changes, which may lead to enhanced perceptual functioning and social impairments indicative of ASC.

As reviewed by [61] it is difficult to separate the biological characteristics (sex) from the socio-cultural factors (gender) that may lead to an underdiagnosis of autistic females compared to males. For instance, the Autism and Developmental Disabilities Monitoring (ADDM) Network reports that 40% of females with autism have an intellectual disability. In comparison, only 32% of males with autism have an ID when identified through health and school data [62]. However [63] found that when assessing the cognitive functioning of autistic children through standardized assessment, there were no sex/gender differences. This again suggests that females are less likely to be identified as autistic unless they present with higher functional needs. Indeed, while research usually estimates the male-to-female ratio to be 4:1, an infant study looking at siblings of autistic children found the ratio to be 1.65:1 [64], suggesting that when ASC is identified using a high-risk cohort rather than clinical referrals, there is a more even ratio between genders.

It is assumed that individuals carry a gendered view of autism due to the proliferation of male media portrayals of autistic people and popularised research that suggests males are more likely to be autistic. Indeed, the gendered stereotypes of females that are often in direct contrast to autistic characteristics and popularized theoretical accounts of autism as a masculinized condition, few have directly tested this relation. However it is essential to assess the views of the general population as although attitudes towards autism may be improving [65], stigma remains common [66], and individuals' biases towards autism can often contribute to personal and professional challenges for autistic people [67]. This population-based study explores whether gender biases the perception of autism on implicit and explicit measures.

## The current study

To measure how gender colours the perception of autism at the subconscious level, an Implicit Association Test (IAT) was employed to measure participants' implicit gendered autism associations by recording their reaction times when categorizing male and female names with autistic traits to observe potential implicit bias between the two categories. It was hypothesized that the association between autistic characteristics would be stronger with males than females in an IAT. Autism Spectrum Quotient (AQ) scores in relation to a male or female vignette were compared through a MANOVA to explore whether certain traits on the AQ will be more heavily endorsed when the AQ is being answered about a hypothetical male than a female.

## Methods

### Design & participants

This study employed a mixed design comparing Neurotypical/ASC—female/male implicit associations within participants and explicit endorsement of fe/male targets on an AQ between participants. A power analysis using G*Power determined that 300 participants would be needed to detect a small effect size of Cohen's $d$ = 0.3. This sample size and hypothesis and analysis plans were pre-registered prior to data collection (https://aspredicted.org/blind.php?x=8xn4v6). Some additional analyses were undertaken and are identified as such.

Three-hundred participants (149 male, 151 female, age range 18 to 72 years, $M$ = 26.79 $SD$ = 10.11) were recruited by opportunistic sampling method through study advertisements

on both online research (survey swap, survey circle) and social media platforms (Reddit, Facebook). Participants' ethnicity was as follows: 81.27% White, 2.22% Black or African American, 12.70% Asian, and 3.81% Other. Participants' country of residence was as follows: 64.00% UK, 9.00% USA, 27.00% Other (i.e., a mixture of 43 countries with less than 3%). All participants gave full informed consent and were required to be above 18 years of age. This study received ethical approval from Edge Hill University's ethical review board. Of the 300 participants, 14 possessed a medical diagnosis of autism (of which 8 completed the female condition and 6 the male condition).

## Materials & procedure

Implicit associations were assessed using an IAT employed on Qualtrics (Provo, UT) study platform. Specifically, participants were presented with a target (male/female names) and attribute (autistic and non-autistic traits) words, one at a time, in the centre of the screen. They were instructed to press 'E' on the keyboard when the picture or word matched the target variable on the left and 'I' when the term matched the target variable on the right. On crucial trials, male/autistic and female/non-autistic words were categorized using the same key. In the remaining trials, the pairings of attributes and targets were switched. Incorrect responses presented participants with a black screen and a red 'X'; they were then prompted to press the correct key. The speed at which participants accurately categorized the words and the number of errors was recorded. Reaction time and accuracy in response to each attribute pairing showed a stronger implicit association between the two items.

A 7-block IAT was used, with each block consisting of 20 trials [68], and in line with recommendations, all trials with reaction times under 300 ms were removed, errors made within each trial were replaced with a participant's mean reaction time for that block, plus an error penalty of 600 ms [69]. Participants' mean reaction time for each association pairing, male autistic/female non-autistic and female autistic/male non-autistic, were then calculated. This produced a total reaction time of both the female autism pairing and male autism pairing, which accounted for both reaction times and errors.

IAT stimuli comprised eight autistic, eight neurotypical (non-ASC) traits, and eight male and eight female names. The names used within the IAT were chosen from amongst the top 100 baby names in the 2000s due to the projected age of most participants completing the study [70]. Names were also chosen based on their phonological similarity, favouring names of a shorter length and fewer syllables [which has been shown to reduce error and increase participant response times in the IAT [71]. The ASC/NT items were chosen based on previous research which used a similar IAT to investigate implicit racial biases and how they influenced autism recognition [72]. In keeping with the recommendations that IAT stimuli be short, some original items were reduced in their word count. For example, 'displays rigid routines' was shortened to simply 'rigid.'

Further IAT and autism research provided additional items for the autism category, such as 'introverted', 'autistic', 'atypical' and 'different.' Neurotypical items were identified as 'extraverted', 'flexible', 'typical', 'independent' and 'social' [73]. Pilot research supports these choices of words, as students were asked what behaviours they think are characteristic of an autistic individual [73]. Broader research on autism also suggests that autistic individuals struggle with social reciprocity and expressing emotion [1, 74], supporting the 'rigid,' 'introverted,' 'reserved,' and 'different' items. IATs require closely related items to comprise the two stimulus categories to make it easier for participants to respond differentially [75]. Therefore antonyms of the autistic word items were used for the words in the non-autistic category, such as introverted (autistic) and extraverted (non-autistic). A complete list of all items used can be found in Table 1.

**Table 1. Representation of IAT categories.**

| Category | Items |
|---|---|
| Male | Josh, Jack, Tom, James, Dan, Ben, Luke, Ryan |
| Female | Emma, Chloe, Sarah, Kate, Amy, Lisa, Emily, Zoe |
| Autistic | Autistic, Reserved, Rigid, Asperger's, Atypical, Different, Logical, Introverted |
| Non-Autistic | Neurotypical, Spontaneous, Flexible, Typical, Extraverted, Social, Talkative, Outgoing |

In addition to the IAT, the 50-item AQ [76] and an accompanying vignette were used to measure explicit associations between certain autistic traits and male and female genders. The AQ is a screening tool devised to measure the degree of autistic qualities an individual may possess. The AQ provides participants with 50 statements, which they must rate in terms of how much they agree or disagree with the statement using a 4-point Likert-type scale ranging from 1 (strongly disagree) to 4 (strongly agree).

The AQ scores range from 0 to 50, with a higher score relating to a higher degree of autistic traits through both social ("I prefer to do things with others rather than on my own") and non-social ("I am fascinated by dates") aspects of cognition and behaviour. Half of the items are reverse-scored. The present study implemented the AQ to assess how participants perceived either a male or female individual described in a preceding vignette. Images were also provided alongside both vignettes, showing a male or female face. These pictures were obtained from the Karolinska Directed Emotional Faces (KDEF) [77] to further prime the gender of the character in the vignette.

The vignettes described an individual with autistic traits and were designed based on overarching autistic characteristics identified in previous research which developed vignettes through an iterative process of consulting autistic adults, clinicians, researchers and educators [42]. The outcome of this research provided the following autistic traits: difficulty socializing, restrictive interests and problems with change. Immediately after completing the IAT, participants read the following vignette (half in the male condition/half female). After that, they responded to the AQ-50 statements as if they were the individual described in that vignette.

"*Matthew/Kerry* is very reserved. *He/she* struggles to make new friends as *he/she* prefers to speak to people *he/she* is familiar and already knows. *He/she* doesn't like it when *his/her* desk is rearranged at work and likes having a routine. At work, *Matthew/Kerry* has difficulty determining if *his/her* boss is happy with *his/her* performance, and he/she often struggles to understand *his/her* colleague's jokes. *He/she* prefers speaking about topics he/she knows a lot about. In *his/her* leisure time, *Matthew/Kerry* plays individual games and sports, like golf, rather than playing games in teams."

Finally, participants were asked to rate their pre-existing knowledge of autism using a 5-point Likert scale (1 –Not at all; 2 –A little; 3 –A moderate amount; 4 –A lot; 5 –A great deal).

## Results

A pre-registered Wilcoxon signed rank test was conducted to test the difference in RTs between the female-ASC pairing and the male-ASC pairing as data was not normally distributed (KS(300) = .16, $p < .001$ for female-ASC pairing and KS(300) = .12, $p < .001$ for male-ASC pairing). Response times were significantly lower when autistic traits were being categorized with male names (median = 972.90, IQR = 241.70) compared to female names

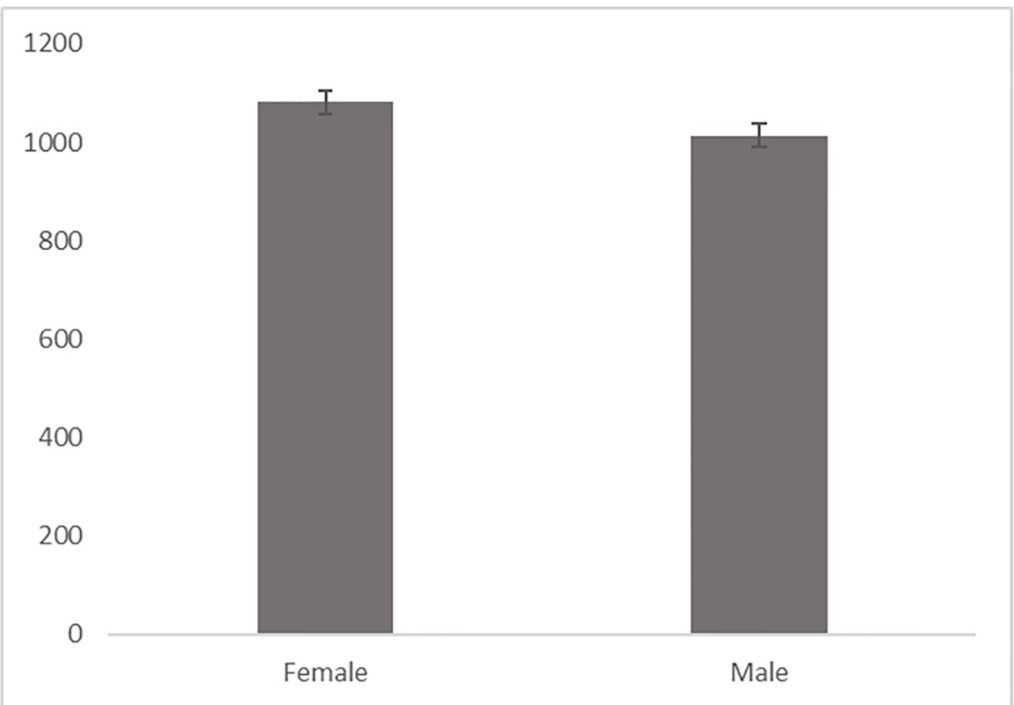

**Fig 1. Mean and standard errors of the mean reaction times for the male and female category.**

(median = 1009.00, IQR = 322.700), U = 26623.00, $p$ = .007, indicating that participants were faster to associate autistic words with male names (please see Fig 1 for mean and standard errors of the reaction times for the male and female category).

Exploratory research was conducted to ascertain whether implicit associations were related to the level of autism knowledge. A Spearman's correlation tested a possible correlation between participants' knowledge of autism and their reaction times. No correlation was found in the male-ASC condition ($r_s$ = -.03, $p$ = .65) or in the female-ASC condition ($r_s$ = -.02, $p$ = .72), suggesting that knowledge of ASC did not influence the reaction times of participants.

A pre-registered between-subjects MANOVA was then carried out, testing the main effect of the vignette gender [male or female condition] on the level of perceived autistic traits assigned to the individual described in the vignette. This MANOVA was not significant (F (249.00) = 454.697, p = .489). However, it was noted that the in the between-subjects effects, one AQ item differed significantly across the conditions below the .05 alpha level, and a further three were also cited as differing below the .10 alpha level. Please see Table 2 for descriptive and inferential. All other items were comparable across conditions (p's >.10).

## Discussion

The present study investigated whether those in a general population sample displayed a male gender bias in their perceptions of autism. A significant implicit gender bias towards associating males with autistic traits was found using the IAT. This implicit association is consistent with a higher male-to-female autism ratio and evidence that autistic females are more likely to remain undiagnosed [7] or be diagnosed much later [9]. This implicit bias towards males and autistic traits may have a detrimental effect on the diagnosis rate of females and have profound effects on their wellbeing [6] and exacerbate masking, making ASC females less likely to

**Table 2. MANOVA results for pertinent AQ-50 items.**

|  | *M* | | *SD* | | *F* values | *P* values | $\eta_p^2$ |
|---|---|---|---|---|---|---|---|
|  | Male | Female | Male | Female |  |  |  |
| 1. I prefer to do things with others rather than on my own. | 2.67 | 2.99 | 1.22 | 1.20 | 5.238 | .023 | .017 |
| 34. I enjoy doing things spontaneously | 3.36 | 3.52 | 0.82 | 0.75 | 3.014 | .084 | .010 |
| 39. People often tell me that I keep going on about the same thing. | 3.24 | 3.06 | 0.84 | 0.78 | 3.709 | .055 | .012 |
| 41. I like to collect information about categories of things [e.g., types of birds, cars, trains, plants]. | 3.24 | 3.08 | 0.85 | 0.82 | 2.769 | .097 | .009 |

receive clinical support [54]. The development of sexual identity may also be affected by the heavily gendered view of autism. Research identifies a key emerging theme of gender dysphoria and a conflict surrounding sexual identity within autistic individuals, gender dysphoric autistic participants have expressed how 'gender-loaded stereotypes' surrounding autism complicated their understanding of their gender [11].

This issue is further highlighted in qualitative research which reports females with ASC experience feelings of 'not fitting in' with society, as their atypical behaviours did not comply with their peers and remained unaccounted for by a lack of a diagnosis of autism [7, 10]. The concept of gender was also frequently discussed in both studies, explicitly mentioning the pressure placed on females 'to be a certain way'–highlighting the detrimental effects of gender bias on autistic female wellbeing. For instance, being quiet and passive within a group was seen as more socially acceptable for females than males, leading to their symptoms' remaining hidden [7]. Overall, suggesting that males with ASC experience less pressure to camouflage their symptoms were a critical feature of both studies. Indeed, females with ASC recognize this as contributing to the difference in diagnosis rates between males and females [10].

This study revealed that the language used to describe ASC might be incongruent with certain gender expectations for females. As shown in the IAT and specific items of the AQ-50, the general view that females pose more emotional traits, alongside greater motivation to communicate for social reasons than males, appears to stand in direct contrast to features associated with autism [23]. The current study provides quantitative support by revealing gender biases towards males and autism. It identifies the significant differences in people's implicit bias towards males and ASC compared to females, alongside evidence of explicit views towards females, with ASC being perceived as more social than males with ASC.

Using a widely used early screener for adults seeking an autism diagnosis, the AQ-50, an explicit bias was demonstrated in the current study, supporting the notion that certain items on the measure are more indicative of male versus female traits. Results from Question 1 on the AQ-50 ("I prefer to do things with others rather than on my own") indicate that females were perceived as more likely to be interdependent than men whilst question 34 ("I enjoy doing things spontaneously") also highlights that females are perceived as more spontaneous than males. Indeed, no autistic items in the AQ-50 were rated more highly for the female vignette than the male vignette. This suggests an explanation for why girls and women often need to present with more concurrent behavioural, developmental or mental health issues for an autism diagnosis would be made, compared to their male counterparts in the current diagnostic practice [52, 54]; thus contributing to females with ASC remaining undiagnosed [5]. It also suggests that commonly used self-report measures are not representative of female autistic traits that may be differentiable from male autistic traits [78].

Similarly, question 39 ("people often tell me that I keep going on and on about the same thing") was rated higher for males. This is consistent with the gender bias that females are perceived to have more developed social skills and can intuit social cues more adeptly than men.

This may explain females' increased ability to camouflage their ASC traits [79] and their tendency to display more socially acceptable restricted interests [80]. Camouflaging is also believed to be increased by the heightened social pressures and expectations that women face compared to males in everyday life [6, 10, 18]. Females may be more likely to mask their traits to fit in more within society and gain a greater understanding and acceptance from their peers due to a perceived dissociation between their gender and diagnosis.

Increased motivation to camouflage due to societal pressures [36] can contribute to the later medical diagnosis of autism received by females compared to males, as gendered views towards autism and males may implicitly influence clinicians. This may place ASC females at increased risk of developing anxiety and depression [6, 81]. A gendered view of autism may pose severe consequences for the wellbeing of females with ASC, putting them at a disadvantage to males [6].

Question 41 ("I like to collect information about categories of things, e.g., types of cars, birds, trains, plants") was rated more highly for the male vignette. This perception may be due to the categories suggested by the question itself (such as cars and trains). These items may be more likely to represent male interests which may automatically lead to more male-dominated assumptions about ASC. For instance, girls with ASC have been found to present with different restricted interests than boys [34], such as high interest in a specific television character/celebrity. In contrast, boys' interests early in development commonly revolve around wheeled toys [82].

One study [83] found autistic children tend to have special interests that fit traditional gender stereotypes (such as sports and Legos for boys and fantasy for girls). This suggests that societal expectations and bias also affect the restricted interests of an individual with ASC. The restricted interests found in boys and girls followed interests that are arguably typical of their gender [82]. These results suggest that socially constructed gender biases shape how ASC-related behaviours are perceived [8], highlighting the importance of changing attitudes towards females presenting with autism. Reducing gender bias towards autistic females would improve the social and developmental outcomes for females with ASC while reducing their likelihood of developing mental health issues.

Since the development of the AQ-50, understanding autism and its presentation in both boys and girls has grown considerably, leading to the development of the Comprehensive Autism Trait Inventory (CATI) [84]. This new measure of autistic traits overcomes shortcomings within the AQ-50 regarding social camouflage and sensory sensitivity, thus providing a more comprehensive self-report measure. The CATI may therefore be a more sensitive measure towards identifying ASC in females as opposed to the AQ-50, however, it is not yet intended to be a diagnostic tool.

There were limitations to this study that could be addressed by further research. First, this study was conducted with an online sample of individuals from the general population. While it was interesting to understand how the general public genders autistic individuals regarding community perceptions and acceptance of autistic people, there are other groups for whom a gendered bias towards autistic males rather than females would be particularly detrimental. Replicating this study with autism professionals is a necessary extension of this work. Understanding male versus female biases in healthcare providers, educators, and clinicians is of specific interest. Such individuals are often the first to identify individuals who may qualify for a diagnosis. It would also be interesting to investigate gender biases among autism researchers, an important consideration given that autism studies have a history of recruiting predominantly male samples. As autism researchers are responsible for creating the measures and interventions widely used to assist autistic individuals, understanding whether there are

systematic biases that colour such tests and programs is an essential next step in improving these shortcomings.

Additionally, this study used a vignette that was interchangeably represented a hypothetical male and female. While vignettes are often used in research to understand social biases by describing overt behaviours, they cannot show the more subtle behaviours often associated with ASC, including verbal and non-verbal communication differences. Video recordings of autistic males and females used to uncover gender biases towards the perception and acceptance of autistic males and females would be appropriate in future. As our research suggests that autistic traits conflict with commonly held stereotypes regarding female versus male social characteristics, it may be that more sensitive stimuli may lead to even greater discrimination against autistic females. This would help explain the higher prevalence of masking in autistic females. Second, while the same vignette was used to compare male and female conditions, it could be argued that it was more representative of male autistic traits, as this study suggests that what is understood to be an 'autistic' trait is itself male-biased. Future research may want to replicate this work with vignettes that showcase more typically female autistic characteristics; for instance, [85] found stereotyped interests and repetitive behaviours to better predict ASC in girls. Such a vignette may reveal even more pronounced discrepancies between males and females regarding popular measures such as the AQ, leading to restructuring existing standards or creating more balanced tests.

In conclusion, the results of this study highlight the effects of gender bias and gender-based expectations on the perceptions of ASC. This adds to the recent subdiscipline of autism research aimed at elucidating the nature of sex and gender differences on the autism spectrum and how this impacts autistic girls and women [8]. The current research provides further evidence that the general public holds gendered views of autism and that specific measures of autistic traits are biased towards males over females. Understanding how this bias extends to autism professionals who are particularly critical in identifying and supporting autistic individuals is paramount, as revealing such a bias could help explain the discrepancies between autistic male and female ratios. Research also suggests that autistic people are significantly more likely to experience gender dysphoric perceptions than neurotypicals. As gendered roles are learned through interaction with societal norms, and this research suggests that society views autism as a male rather than a female manifestation, it would be interesting to examine how this contributes to gender dysphoria in autistic individuals. This research suggests that current popular perceptions of autism are conflated with male rather than female stereotypes, and widely used measures such as the AQ present items meant to measure autistic traits that are simultaneously congruent with male rather than female stereotypes. Future research should include items in measures such as the AQ that align with common autistic female characteristics and behaviours or create a new version of such measures geared explicitly towards females. Finally, while this study identified a significant pattern of male versus female bias towards ASC traits, it did not elucidate why this pattern exists, though several hypotheses were suggested. Understanding why general perceptions of autism are skewed towards males rather than females and educating individuals on female autism presentation is paramount to improving the identification and widespread acceptance of autistic females in the broader community.

## Author Contributions

**Conceptualization:** Liam Cross.

**Data curation:** Rae Brickhill.

**Formal analysis:** Rae Brickhill.

**Supervision:** Gray Atherton, Liam Cross.

**Writing – original draft:** Rae Brickhill.

**Writing – review & editing:** Gray Atherton, Andrea Piovesan, Liam Cross.

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
