## [Decision Letter · Decision Letter 0]

10 Jan 2023

PONE-D-22-00876Autism, thy name is man: Exploring implicit and explicit gender bias in autism perceptionsPLOS ONE

Dear Dr. Brickhill,

Thank you for submitting your manuscript to PLOS ONE. After careful consideration, we feel that it has merit but does not fully meet PLOS ONE’s publication criteria as it currently stands. Therefore, we invite you to submit a revised version of the manuscript that addresses the points raised during the review process.

We look forward to receiving your revised manuscript.

Kind regards,

Asem Surindro Singh, Ph.D

Academic Editor

PLOS ONE

Journal Requirements:

Additional Editor Comments (if provided):

I thank the authors for submitting the manuscript entitled "Autism, thy name is man: Exploring implicit and explicit gender bias in autism perceptions" to PLOS ONE. Authors claimed that there is bias in the diagnosis of autism between males and females that leads to higher number of autisms occurred among males compared to females. This article will help in providing equal importance between males and females in the diagnosis of autism, if there exists a bias or chances of bias that may occur in the future.

Authors are requested to answer the comments of the reviewers and reflect the concerns in the manuscript.

From the genetic point of views there are many articles that reported higher number of males with autism than females, thereby more chances of occurring autism among males than females. I would request the authors to address this point as well.

We look forward to receiving the revised manuscript.

Yours Sincerely

Asem

Reviewers' comments:

Reviewer's Responses to Questions

**Comments to the Author**

1. Is the manuscript technically sound, and do the data support the conclusions?

Reviewer #1: Partly

Reviewer #2: Yes

2. Has the statistical analysis been performed appropriately and rigorously? 

Reviewer #1: Yes

Reviewer #2: I Don't Know

3. Have the authors made all data underlying the findings in their manuscript fully available?

Reviewer #1: No

Reviewer #2: No

4. Is the manuscript presented in an intelligible fashion and written in standard English?

Reviewer #1: Yes

Reviewer #2: Yes

5. Review Comments to the Author

Reviewer #1: PONE-D-22-00876: Brickhill & al., "Autism, thy name is man: Exploring implicit and explicit gender bias in autism perceptions"

This manuscript reports a significant implicit association between autistic trait words and male names in contrast to female names, and also identifies three or four (depending on criterion) items in the fifty-item version of the Autism Spectrum Quotient (AQ) whose ratings by a general-population sample are biased by the gender of the subject, and argues that, therefore, the AQ is a gender-biased instrument resulting in gender-biased autism diagnostic screening.

The recent publications most relevant to this argument, those introducing the unbiased Comprehensive Autistic Trait Inventory (English & al., 2021, doi: 10.1186/s13229-021-00445-7) and detailing the differences in categorical diagnoses between girls and boys with similar genetic loadings for autism (Burrows & al., 2022, doi: 10.1016/j.biopsych.2022.05.027), are not mentioned.

The culture (or cultures) from which survey respondents are drawn is not specified. This detail is important because gender norms vary across cultures.

In the Implicit Association Test, an error penalty of 600 ms seems arbitrary and liable to skew averages, and both this penalty and the replacement with the current block's mean reaction time seem liable to skew means and variances. Why not just omit error trials from the computation of mean reaction time?

In the report of mean reaction times by condition, are the units milliseconds (ms)? Specify. And what is the proper number of significant figures; can the hardware actually measure intervals as short as tenths of microseconds? Typical USB keyboard scan intervals are on the order of a millisecond.

The text reports "One AQ item was identified as differing significantly across the conditions at the .05 alpha level, and a further 3 at the .01 alpha level" but the tail probabilities in Table 2 do not conform to this description, and the data for AQ item #39 in particular seem not to argue at all strongly against the null hypothesis. Also in Table 2, it's unclear why the Bayes factor is missing in the first (AQ item #1) data row, the narrow column widths cause data to be broken across rows, and it might be simpler to report a single Bayes factor than to report a pair of reciprocals.

Statements of interpretation such as "participants found it easier to associate autistic words with male names" and "These empirical findings reveal a tendency in the average person to associate autistic traits with males compared to females" should be confined to the discussion section.

The manuscript file asserts "The data that support the findings of this study are available on request from the corresponding author. The data are not publicly available due to privacy or ethical restrictions." Yet the "Data Availability" question is answered "Yes - all data are fully available without restriction". This contradictory pair of responses violates the instruction "Important: Stating `data available on request from the author' is not sufficient. If your data are only available upon request, select `No' for the first question and explain your exceptional situation in the text box."

I have annotated the authors' manuscript with suggested (tracked) changes and marginal notes for the authors' use in revision, and am uploading this anonymously annotated manuscript as part of this review; please see this annotated file is forwarded to the authors.

Reviewer #2: WHAT IS THE STUDY ABOUT? Female stereotyped behaviour stands in stark contrast to typically recognised autistic traits. Women display more emotional expression and communication in social situations, thereby masking diagnosis and delaying intervention. Abstract and Introduction handle the above summary. Abbreviations should include the expanded form.

METHOD: Population based study appropriate. Sample size of 300 selected by appropriate power analysis. Cross cultural, online and social media search however needed further elaboration on the methodology of collecting data. Age range mentioned. Ethical approval taken. Informed consent taken. There was no clarification about the participations previous knowledge of Autism which could influence the results.

RESULTS: Statistical Analysis not checked: needs review by statistician.

Results support conclusions. Limitations have been discussed. The study is on a valid topic and very essential for general public understanding to remove stereotyped assumptions. Important also for health sector, education and researchers. This study shows the need for early diagnosis and will help to reduce the demand on females with autism for higher level of functional needs in behaviour, development, thus affecting mental health. The author has also mentioned objection about the Male brain term.

CONCLUSION: Data is enough to recreate the analysis of the selected points by the author. Previous knowledge on autism is necessary to give a correct picture.

RECOMMENDED CORRECTIONS: "Greater communality'' to be replaced by "greater communication ability"

Needs correction "Non autistic characteristic are talkative". High functioning verbal women are also talkative in their own interest area. This criterion needs modification.

Abbreviations need to be given in expanded form at the initial stage.

PLAGIARISM: Not checked

6. PLOS authors have the option to publish the peer review history of their article (what does this mean?). If published, this will include your full peer review and any attached files.

Reviewer #1: **Yes: **Matthew K. Belmonte

Reviewer #2: **Yes: **Dr Shabina Ahmed

---

## [Author Response · Author response to Decision Letter 0]

10 Feb 2023

We thank the reviews for their rigorous comments and corrections, these changed have now all been adhered to. Specific reviewer and editor comments are addressed in the 'Response to viewers' file attached.

---

## [Editor Report · Decision Letter 1]

6 Mar 2023

PONE-D-22-00876R1Autism, thy name is man: Exploring implicit and explicit gender bias in autism perceptionsPLOS ONE

Dear Dr. Atherton,

Thank you for submitting your manuscript to PLOS ONE. After careful consideration, we feel that it has merit but does not fully meet PLOS ONE’s publication criteria as it currently stands. Therefore, we invite you to submit a revised version of the manuscript that addresses the points raised during the review process.

Authors have carefully revised the manuscript and is appreciated. I would like to add an additional comment to the authors which will be benefited to the readers. There are many reports from the genetic studies, that indicated overrepresentation of autism in males compare to females are likely due to genetic factors. Subsequently, if there is sex linked genes, logically, overrepresentation of autism in males makes sense. I would appreciate if authors could briefly address this point as well. 

We look forward to receiving your revised manuscript.

Kind regards,

Asem Surindro Singh, Ph.D

Academic Editor

PLOS ONE
---

## [Author Response · Author response to Decision Letter 1]

8 Mar 2023

Authors have carefully revised the manuscript and is appreciated. I would like to add an additional comment to the authors which will be benefited to the readers. There are many reports from the genetic studies, that indicated overrepresentation of autism in males compare to females are likely due to genetic factors. Subsequently, if there is sex linked genes, logically, overrepresentation of autism in males makes sense. I would appreciate if authors could briefly address this point as well. 

Thank you very much for this comment. You are absolutely right, this is indeed an important area of research that may indeed inform public opinion. I have included two additional prominent theories of why there may be a female protective factor with regards to genetic liability for autism, and why males may be more likely to develop autism using theories relating to genetic differences sex wise in neural plasiticity. I have also included research that suggests that if females are diagnosed younger and using standardised assessments the sex ratio is less pronounced and females have no differences in cognitive functioning compared to males. This suggests that there is a complex relationship between gender, sex and autism that may not be clear with regards to genetic liability. I hope this makes sense and thank you for this comment.

---

## [Editor Report · Decision Letter 2]

22 Mar 2023

Autism, thy name is man: Exploring implicit and explicit gender bias in autism perceptions

PONE-D-22-00876R2

Dear Dr. Grey Atherton,

We’re pleased to inform you that your manuscript has been judged scientifically suitable for publication and will be formally accepted for publication once it meets all outstanding technical requirements.

Kind regards,

Asem Surindro Singh, Ph.D

Academic Editor

PLOS ONE

---

## [Editor Report · Acceptance letter]

2 Aug 2023

PONE-D-22-00876R2 

Autism, thy name is man: Exploring implicit and explicit gender bias in autism perceptions 

Dear Dr. Atherton:

I'm pleased to inform you that your manuscript has been deemed suitable for publication in PLOS ONE. Congratulations! Your manuscript is now with our production department. 

Kind regards, 

on behalf of

Dr. Asem Surindro Singh 

Academic Editor

PLOS ONE